# Genome-Wide Search for Associations with Meat Production Parameters in Karachaevsky Sheep Breed Using the Illumina BeadChip 600 K

**DOI:** 10.3390/genes14061288

**Published:** 2023-06-19

**Authors:** Alexander Krivoruchko, Andrey Likhovid, Anastasiya Kanibolotskaya, Tatiana Saprikina, Elena Safaryan, Olesya Yatsyk

**Affiliations:** 1Federal Seфey Budgetary Scientific Institution, North Caucasian Federal Scientific Agrarian Centre, 356241 Mikhailovsk, Russia; dorohin.2012@inbox.ru (A.K.); saprikina.tanya@mail.ru (T.S.); telegina.helen@yandex.ru (E.S.); malteze@mail.ru (O.Y.); 2Department of Genetic and Selection, FSAEIHE, North-Caucasus Federal University, 355017 Stavropol, Russia; alikhovid@ncfu.ru

**Keywords:** sheep, SNP, genome-wide association search, GWAS, candidate gene, Karachaevsky sheep breed

## Abstract

In a group of Karachaevsky rams, a genome-wide associations analysis of single nucleotide polymorphisms (SNPs) with live parameters of meat production was performed. We used for genotyping the Ovine Infinium HD BeadChip 600 K, which consists of points to detection of 606,000 polymorphisms. A total of 12 SNPs was found to be significantly associated with live meat quality parameters of the corpus and legs and ultrasonic traits. In this case, 11 candidate genes were described, the polymorphic variants of which can change in sheep body parameters. We found SNPs in the exons, introns, and other regions of some genes and transcripts: *CLVS1*, *EVC2*, *KIF13B*, *ENSOART00000000511.1*, *KCNH5*, *NEDD4*, *LUZP2*, *MREG*, *KRT20*, *KRT23* and *FZD6*. The described genes involved in the metabolic pathways of cell differentiation, proliferation and apoptosis are connected with the regulation of the gastrointestinal, immune and nervous systems. In known productivity genes (MSTN, MEF2B, FABP4, etc.), loci were not found to be a significant presence of influence on the meat productivity of the Karachaevsky sheep phenotypes. Our study confirms the possible involvement of the identified candidate genes in the formation of the phenotypes of productivity traits in sheep and indicates the need for new research into candidate genes structure in point to detect their polymorphisms.

## 1. Introduction

The use of genetic technologies has ensured stable progress in breeding work to improve the productive qualities of farm animals. Due to the use of molecular genetic markers, it has become possible to predict the productivity of animals and select parental pairs for crossing taking into account genotypes. This significantly reduces the time for improving the breeding characteristics of existing breeds and producing new breeds of farm animals. Therefore, the study of the influence of various genes on productive traits has been and remains the most important task for fundamental and applied genetics in animal husbandry [1]. As a result of the conducted studies, genes associated with the meat [2,3] and dairy productivity of sheep [1,4] were identified.

The parameters of meat productivity in sheep are determined by the intensity of muscle tissue growth due to the proliferation and differentiation of myocytes. This process is controlled by a large number of genes, among which there are both agonists and antagonists of muscle growth. The mechanism of action of some of them has already been well studied, and the structural features of these genes leading to changes in the phenotype of animals have also been described. The effect of the myostatin gene on the development of muscle tissue, expressed in the inhibition of muscle growth in mammals, has been most studied. Mutations in this gene are described, which are accompanied by a violation of its function and an increase in muscle mass in animals [5,6]. Mutations in the Callipygian locus are also accompanied by increased muscle growth in the thigh region in farm animals [7]. Among the agonists of myocyte proliferation, *Follistatin* and *MYOD1* genes are the most studied; their function is realized through the influence on myostatin expression [8,9,10]. The identification of polymorphisms in the described genes is already widely used in marker-associated breeding in sheep to increase meat productivity.

The introduction of genetic technologies into the complex of breeding measures requires an assessment of the impact on the productive qualities of as many genes as possible. Therefore, the search for new candidate genes that affect the meat productivity of sheep is one of the most important tasks in animal genetics. The main method of identifying such genes is the genome-wide association search (GWAS) of as many genome loci with the studied trait as possible. It is performed both using genome-wide sequencing of a new generation and using DNA biochips mapping single nucleotide polymorphisms (SNPs). In recent years, research has been conducted to find new candidate genes in sheep of various breeds. As a result, a number of candidate genes has been proposed as being related to sheep meat productivity [11,12], shearing and wool quality [13] and dairy qualities [14]. The functions of the genes identified in the course of the work relate to the regulation of various processes in the nervous and muscular tissue, coding of proteins and enzymes of energy metabolism, as well as participation in the work of the immune system [11,13]. However, given the complexity and multifactorial nature of the network of genes that determine the state of muscle tissue, the search for new candidate genes continues to be an urgent task in animal husbandry.

The identification of new candidate genes in endemic sheep breeds that are bred in regions isolated due to geographical conditions is of particular interest. Due to such genetic isolation, it is possible to stabilize the effect on the productivity of some genes and the manifestation of the influence of others [15]. One such endemic breed is the Karachaevsky breed, bred in the foothills of the North Caucasus since the early 1800s and which since then has not had direct contact with other breeds. The researchers noted the independent origin of the breed by improving local sheep and the absence of genetic links with modern sheep breeds. The breed is well adapted for breeding in year-round grazing conditions. Today, the farms of Karachay-Cherkessia contain more than 250,000 heads of this breed. Rams of the Karachaevsky breed have a live weight of 60–70 kg and the queens reach 50 kg. The wool of the animals is coarse, but the breed is used as a universal one, receiving meat and milk from sheep. They ripen quickly for animal meat, have a slaughter yield of more than 55% and produce delicious meat with a rich taste. Milk productivity is up to 50 kg of commercial milk per lactation. The fat content of the milk is on average 9.6%. Rams produce about 3 kg of wool per haircut while queens produce more than 2 kg. Due to their habitat in foothill areas with a changeable climate, animals are resistant to adverse conditions and rarely suffer from infectious diseases [16,17].

In the course of our research, we conducted genotyping of sheep of the endemic Karachaevsky breed using the Illumina Ovine Infinium HD BeadChip 600 K. A GWAS for SNPs and meat productivity traits was performed. As a result, polymorphisms were identified that were reliably associated with live parameters of meat production. Analysis of the localization of these polymorphisms in the genome allowed us to propose a number of new candidate genes, the structure of which may be related to the productive qualities of animals.

## 2. Materials and Methods

### 2.1. Ethics Statement

The sample collection and study purpose were approved by the Institutional Animal Care and Use Committee (approval number 2022–0016, 14 February 2022) of the All-Russian Research Institute of Sheep and Goat Breeding, Stavropol, Russian Federation.

### 2.2. Animals Used and Obtaining Samples

The rams of the Karachaevsky sheep breed at 12 months were this study object. Maximal pure breed quality rams (n = 276) of one herd in the North Caucasus Territory (Russian Federation) were genotyped. In a group of Karachaevsky rams, the live meat productivity parameters were measured. The live weight at birth and at 12 months was measured using scales, and the daily weight gain was calculated from these parameters. The height at the withers and croup, the width of the chest and back, the depth of the chest, and the girth of the shoulder, forearm, and thigh were measured with a grading ruler and measuring tape [18]. The parameters of the depth and width of the muscle “eye”, the depth of the femoral muscle, and the adipose tissue in the lumbar region were determined via ultrasound [19]. All selected rams were clinically healthy, kept in optimal conditions, and fed with a total mixed ration. From April to September, the sheep were grazing in the alpine meadows of the Caucasus. During the rest of the year, they were kept in the lowlands and fed with hay and concentrates in a ratio of 10:1.

### 2.3. Extraction of DNA and Genotyping

Genomic DNA was isolated from whole blood samples taken under aseptic conditions from the jugular vein using a PureLink Genomic DNA MiniKit (Invitrogen Life Technologies, Carlsbad, CA, USA) in accordance with the manufacturer’s protocol. DNA was isolated from 100 µL of blood. The average concentration was 25 ng/µL. Animal genotyping was performed using the Ovine Infinium HD BeadChip 600 K (Illumina Inc. San Diego, CA, USA) according to the manufacturer’s protocol. Genotyping was carried out in the laboratory of the SCOLTECH, Skolkovo (Moscow, Russia). Initial processing of the genotyping results was performed using Genome Studio 2.0 software (Illumina Inc., San Diego, CA, USA). 

### 2.4. Quality Control of Genotyping

Quality control of genotyping was carried out using PLINK V.1.07 software [20]. The data processing included samples with a call rate of detected SNPs of more than 0.95. Loci with a minor allele frequency (MAF) of less than 0.05 and a missing genotype of more than 0.1 were also excluded. The value *p* = 0.0001 was used as the threshold according to the Hardy–Weinberg equilibrium criterion (HWE). From a total of 276 genotyped animals, 271 samples underwent genotyping quality control and were studied in the second stage. After removing SNPs that failed the frequency test, markers that failed the HWE test and 432 SNPs with unknown positions (Chromosome 0), 508,448 polymorphisms were used for further analysis.

### 2.5. Genetic and Statistical Analysis

The genome-wide association study was performed using the PLINK V.1.07 software [20] for quantitative traits. It is based on the assessment of the significance of SNPs’ influence on quantitative meat production trait variability. SNPs on the X chromosome were not excluded, since the PLINK software allows one to search for associations in males, taking this into account also when controlling the quality of genotyping. Visualization and graphing were performed using a “QQman” package in the programming language “R”. Heatmaps were designed in the Genome Studio 2.0 software (Illumina Inc., San Diego, CA, USA). The search for candidate genes was performed in half of a centimorgan region (250,000 bp upstream and downstream) around the SNPs that showed significant associations with meat production traits. Within this interval, the frequency of crossing over is only 0.5%, and the SNP is highly likely to be inherited together with the adjacent gene. For SNP mapping and alignment, an Ovis_Aries_3.1 genome assembly was used. The description of genes was performed using the genomic browsers UCSC (www.genome.ucsc.edu, (accessed on 14 May 2023) and Ensembl (www.ensembl.org, (accessed on 14 May 2023)). A significant difference was accepted if the *p*-value was lower than 0.01 for an individual SNP and lower 0.00001 for the GWAS.

## 3. Results

A genome-wide associations study of SNPs with a number of lifetime measurements characterizing meat production indicators in sheep of the Karachaevsky breed revealed a fairly large number of polymorphisms (Figure 1). For them, the *p*-value parameter was higher than the threshold value set at −log_10_ (p) = 5. From them, for further research, we selected 12 substitutions with the highest values −log_10_ (p) for four measurements: height at croup, width of chest, width of back and depth of adipose tissue in lumbar region. The most polymorphisms among the selected ones were on chromosomes 2 and 9. The maximum confidence values of the *p*-value were recorded for two SNPs identified during the analysis of associations of the width of chest indicator. The lowest level of confidence among the selected polymorphisms was found in the SNP located on chromosome 21 and associated with the width of back. Polymorphisms located on the X chromosome were included in the association study. However, we did not find SNPs on this chromosome with the required indicators of association reliability for any of the studied phenotypic traits.

The localization analysis of 12 SNPs selected by us showed that seven polymorphisms were located in the intergenic space at a distance of 18,705–259,847 bp from the nearest gene (Table 1). One polymorphism was located in the exon of the protein-coding gene and was a missense variant with an amino acid replacement in the protein structure. Five SNPs were found in the introns of genes, mainly 1–2, but one replacement was in intron 19–20. Allelic variants of substitutions were mostly represented by T and C nucleotides.

The study of the location of polymorphisms relative to nearby genes revealed 12 candidate genes located within half of the centimorganide. One of the genes was located at a distance exceeding the one selected for polymorphisms by only 9874 bp, so we did not exclude it from the study. This SNP showed a reliable relationship with the width of chest parameter and, together with the replacement of rs412409461, was used to annotate candidate genes. Despite the fact that the parameter depth of adipose tissue in the lumbar region had associations with five SNPs and the number of possible candidate genes also turned out to be five, three polymorphisms were located in the intron of one gene. Among them, one was also mapped in the upstream region of another gene, and one polymorphism was located next to two genes at once. A study of the associations of individual SNPs with other measurements characterizing the lifetime parameters of meat productivity in Karachai sheep did not reveal reliable signs of a connection.

Another area of our research was the study of the effect of known genes on meat productivity in Karachaevsky sheep, the connection of which with the development of muscle tissue has already been proven in other breeds.

To perform this, we used data on polymorphisms of loci in which known productivity genes are located, obtained by genotyping on Illumina Ovine Infinium HD BeadChip 600 K. Because polymorphisms in these genes can have the opposite effect on productivity, we selected 50 samples from a group of genotyped animals with similar genotypes at the loci of the studied genes and evaluated the associations of polymorphisms with the two most indicative measurements—live weight and depth of muscle “eye” (Table 2). As a result, it was found that polymorphisms in the loci of the *MSTN*, *MEF2B* and *FAB4* genes in the studied animals have only one homozygous variant of the genotype. Accordingly, it was not possible to detect any effect on productive qualities with such a genotype. For the remaining SNPs, we did not find reliable associative links with the studied parameters of meat productivity.

The heat map (Figure 2), constructed on the basis of genotype data for the studied loci in the selected group of animals, showed that only homozygous variants (*MSTN*, *MEF2B*, *FABP4*), predominantly mutant, are present at several loci. 

Most of the loci are represented by all three variants of genotypes—wild and mutant homozygotes, as well as heterozygotes. When clustering a group of animals based on the available genotypes, the absence of any regularity in the distribution of the studied polymorphisms was revealed. At the first and second levels of clustering, no large clusters of genotypes were detected, and at the next levels there was fragmentation into small groups. Thus, despite the variance in the phenotypic parameters of the animals in the sample, a random distribution among the Karachaevsky sheep was observed with respect to genotypes according to known productivity genes. If we were not able to divide animals into two groups based on a complex genotype, then assessment of its effect on the parameters of meat productivity of animals was not performed.

## 4. Discussion

The study of associations of individual SNPs in sheep of the Karachaevsky breed with in vivo parameters of meat productivity revealed several polymorphisms that were significantly associated with the phenotypic characteristics of animals. Based on the data on their location in the genome and the assumption of joint inheritance of a region about half the size of a centimorganide, candidate genes for productive traits located in these regions were annotated.

The rs422045180 replacement is located at a distance of about 125 kbp from the *CLVS1* gene. It encodes the protein clavesin 1, which is necessary for the proper formation of endosomes. It participates in clathrin-mediated endocytosis due to its ability to bind to clathrin and phosphatidylinositol 3,5-bisphosphate [21]. The effect of clavesin 1 on the productive qualities of animals has not been studied. In humans, this protein regulates the intensity of oxidative stress and endocytosis in podocytes. Structural disorders of clavesin 1 are associated with the development of steroid-dependent nephrotic syndrome [22]. Due to the influence of the *CLVS1* gene on the metabolic process, we think it necessary to consider it as a candidate gene for meat productivity in sheep.

Located in exon 24 of the *EVC2* gene, the rs417736092 polymorphism is a missense variant. In its presence, the GCG codon is replaced by GTG, which is accompanied by a change in the amino acid chain of the protein. Thus, the amino acid alanine changes to valine in the composition of the protein ciliary complex subunit 2. It participates in the normal process of bone morphogenesis and skeletal development. There are no data on the effect of the *EVC2* gene in animals. In humans, mutations in this gene are associated with the development of autosomal recessive skeletal dysplasia, also known as chondroectodermal dysplasia. At the same time, there are disproportions in the body and limbs, a shortening of the ribs, violations in the development of teeth and the cardiovascular system [23]. Considering the importance of controlled processes of development of many body systems, we have included the *EVC2* gene in the list of candidate genes for productivity.

The rs413063685 polymorphism is located in the intron of the *KIF13B* gene-encoding kinesin family member 13B. The protein belongs to the group of microtubular kinesin motors. It was found in the primary cilia of mouse fibroblasts and cultured human retinal cells. Disorders of the cellular cilia as a result of mutations of various genes lead to a number of hereditary diseases, united in the class of ciliopathies. Studies have shown that kinesin *KIF13B* ensures the proper functioning of ciliary organelles due to interaction with proteins of the Sonic hedgehog group [24]. Thus, it is involved in a number of important physiological processes and should be considered as a gene that affects the productivity of animals.

On chromosome 10, we found a replacement rs404643411 located far enough away from the coding DNA site. Despite this, we suggest that attention should still be paid to this region, because it contains the RNA recognition motif 1 (RRM1). This motif is found in the RRM domain-containing protein group. When considering biological processes involving proteins containing RNA recognition motif 1, their connection with RNA splicing processes was found due to their participation in the formation of the spliceosome [25]. Due to this, the locus we are considering can influence post-transcriptional changes in various mRNAs and be associated with the realization of the productive qualities of sheep.

Another replacement on chromosome 7, rs412409461, also turned out to be located at a sufficient distance from the nearest coding regions of DNA. The *KCNH5* gene is located almost at the very border of the search interval we have defined. It encodes the potassium voltage-gated channel subfamily H member 5. Proteins of this family are involved in a large number of physiological processes. These include the regulation of neurotransmitter production in neural contacts, hormone release from endocrine cells, and cardiac function [26]. We consider it necessary to evaluate the effect of this gene on the growth and development of animals.

The single nucleotide polymorphism rs430586829 is located near the *NEDD4* gene-encoding NEDD4-like E3 ubiquitin protein ligase. To date, there are no data on the relationship of this group of ligases with the productive qualities of animals. In humans, the functions of NEDD4-like E3 ubiquitin protein ligase are associated with the processes of cell proliferation and differentiation. Changes in the structure of the *NEDD4* gene involved in a number of intracellular mechanisms of nucleartranslation of proteins, lysosomal and proteosomal degradation dramatically increase the risk of developing malignant neoplasms [27]. All this indicates the importance of considering the gene in terms of its impact on productivity, depending on the intensity of muscle tissue growth.

On chromosome 21, we found a replacement rs411231795 located in the intergenic space. The *LUZP2* gene is closest to it and the leucine zipper protein 2 mRNA is transcribed from it. Under normal conditions, this protein is produced in the adult brain and spinal cord at the embryonic stage. When its production is disrupted, a number of severe syndromes associated with the development of tumor processes, genital anomalies, and mental disorders is detected in humans [28]. The importance of this gene for ensuring the normal functioning of the body allows us to consider it a promising candidate gene for productivity in sheep.

The rs420787493 replacement on chromosome 2 is located in the intron of the *MREG* gene. The melanoregulin protein encoded by it has a number of biological functions, mainly related to melatonin metabolism. Its connection with the biogenesis of organelles containing melatonin in cells of various tissues is most indicative. Mutations in the gene can be accompanied by color changes in experiments on mice. However, its influence on the processes of maturation of lysosomes, participation in the release of their contents from cells, connection with the processes of phagocytosis and other functions of membrane formations in the cytoplasm has been shown [29]. We consider *MREG* to be a potential candidate gene for productive qualities in sheep.

In the course of the study, the rs404092476 replacement showed a connection with productive qualities. It was located between two genes, *KRT20* and *KRT23*. Both of these genes encode proteins from the keratin group. Keratins are filamentous proteins that provide intercellular interactions of epithelial cells. They are divided into two large subgroups—cytokeratins and keratins of the hair. Cytokeratins (which include KRT20 and KRT23) are the main cellular proteins of enterocytes and are actively produced in the cells of the stomach and intestines in mammals. The effect of cytokeratin genes on productivity in animals has not been studied. In humans, impaired cytokeratin gene expression is associated with the development of malignant intestinal tumors [30]. The direct connection of the *KRT20* and *KRT23* genes with the metabolism of the epithelium of the gastrointestinal tract allows us to consider them candidate genes for productivity in sheep.

Immediately, three substitutions rs400604822, rs425102210 and rs419561696 were located in the introns of one gene-*FZD6*. Its protein product is a frizzled class receptor 6 from the group of transmembrane receptors for Wnt signaling proteins. Directly, receptor 6 is a negative regulator of the Wnt/β-catenin signaling cascade, thereby inhibiting cell proliferation and apoptosis. Deletion in this gene was found to be associated with the development of autosomal recessive nail dysplasia [31]. *FZD6* has also been implicated in the development of malignant esophageal carcinoma in humans [32]. Thus, the participation of the *FZD6* gene in several important biological processes makes it possible to name it as a gene candidate, affecting productivity.

The known genes of meat productivity during the GWAS did not show a reliable relationship of polymorphisms in their loci with the studied parameters of the exterior in sheep of the Karachaevsky breed. To draw a final conclusion about the possibility of using polymorphisms in these loci for selection, we conducted an additional study of their effect on the studied indicators. In the group of individuals selected from the main pool of genotyped animals with the closest variants of complex genotypes according to the loci of known productivity genes, it was not possible to identify significant associations with the main lifetime signs of meat productivity. Additionally, this conclusion confirms the impossibility of clustering the selected group of animals based on data on complex genotypes. A similar pattern with respect to genotypes by loci of known meat productivity genes was observed in our previous studies in sheep of the Jalgin merino breed [33]. This indicates a certain stability in the influence of known productivity genes created as a result of the development of the breed. Consequently, in sheep of the Karachaevsky breed, the variance in the meat productivity phenotype is associated with other genes that we propose as candidates and recommend for closer study.

## 5. Conclusions

We provided a genome-wide associations study of single nucleotide polymorphisms with lifetime parameters of meat production in selected rams of the Karachaevsky sheep breed. The Ovine Infinium HD BeadChip 600 K was used for genotyping and 606,000 SNPs were detected for all samples. In the result, 12 SNPs were associated significantly with the muscle tissue parameters, characterized by body measurements and ultrasound traits. In this case, 11 candidate genes were described, the polymorphic variants of which can changes in sheep body parameters. We found SNPs in the exons, introns, and other regions of some genes and transcripts: *CLVS1*, *EVC2*, *KIF13B*, *ENSOART00000000511.1*, *KCNH5*, *NEDD4*, *LUZP2*, *MREG*, *KRT20*, *KRT23* and FZD6. The described genes involved in the metabolic pathways of cell differentiation and proliferation, apoptosis are connected with the regulation of the gastrointestinal, immune and nervous systems. Our study confirms the possible involvement of the identified candidate genes in the formation of the phenotypes of productivity traits in sheep and indicates the need for new research into candidate genes structure in direction to detect their polymorphisms. In known productivity genes (MSTN, MEF2B, FABP4, etc.), loci were not found to be a significant presence of influence on the meat productivity of the Karachaevsky sheep phenotypes. 

## Figures and Tables

**Figure 1 genes-14-01288-f001:**
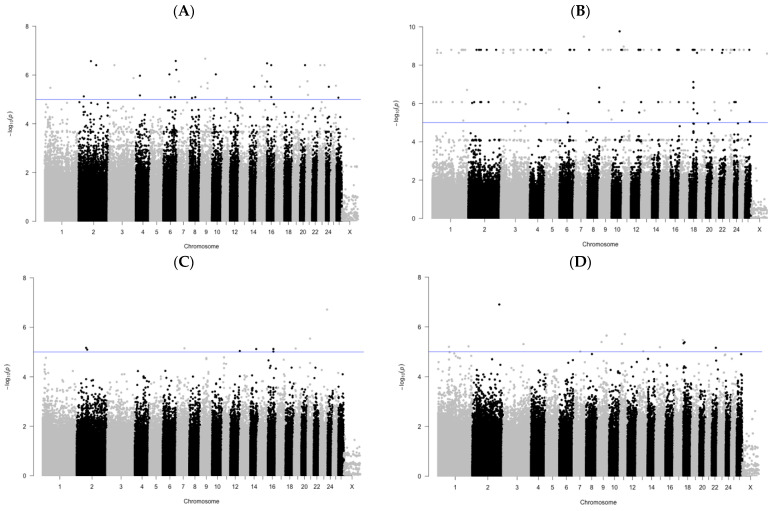
Manhattan plots of GWAS results with a set of −log10 (p) values for the studied SNP associations with meat production traits in Karachaevsky sheep breed. Line indicates the threshold of significance of differences with the value of −log10 (p) = 5. (**A**) Height at croup; (**B**) Width of chest; (**C**) Width of back; (**D**) Depth of adipose tissue in lumbar region.

**Figure 2 genes-14-01288-f002:**
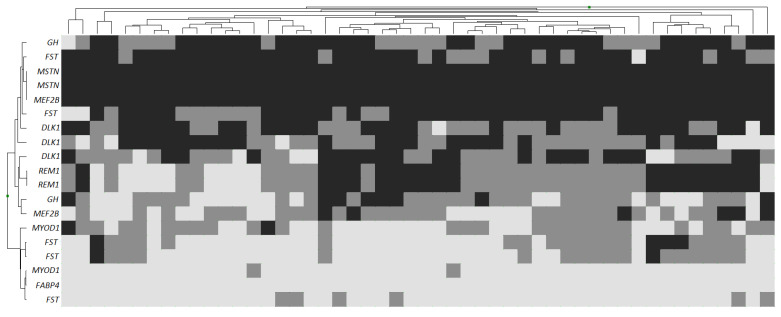
Heatmap of known meat production genes loci polymorphisms in Karachaevsky sheep breed. Clustering by individual genotypes located along the *x*-axis. Light grey—wild homozygote; Grey—heterozygote; Dark grey—mutant homozygote.

**Table 1 genes-14-01288-t001:** Associated-with-meat-production-traits SNPs and candidate genes in Karachaevsky sheep breed.

Trait	Chr	SNP	Position	*p*-Value	Alleles	Gene/Distance, bp
HC	9	rs422045180	39,532,603	2.124 × 10^−7^	T/C	*CLVS1*/124 459
	6	rs417736092	103,289,306	2.648 × 10^−7^	C/T	*EVC2*/exon 24
	2	rs413063685	102,482,150	2.694 × 10^−7^	T/C	*KIF13B*/intron 19–20
WC	10	rs404643411	61,939,091	1.734 × 10^−10^	A/C	*ENSOART00000000511.1*/259 847
	7	rs412409461	71,628,646	3.286 × 10^−10^	C/T	*KCNH5*/244 808
WB	23	rs430586829	57,324,967	1.921 × 10^−7^	G/A	*NEDD4*/88 688
	21	rs411231795	19,797,164	2.875 × 10^−6^	T/C	*LUZP2*/54 924
DAT	2	rs420787493	216,958,135	1.256 × 10^−7^	T/G	*MREG*/intron 2–3
	11	rs404092476	40,665,455	1.961 × 10^−6^	C/A/T	*KRT20*/18 705; *KRT23*/25 186
	9	rs400604822	73,908,694	2.238 × 10^−6^	C/T	*FZD6*/intron 1–2
	9	rs425102210	73,915,521	2.238 × 10^−6^	G/T	*FZD6*/intron 1–2
	9	rs419561696	73,922,648	2.238 × 10^−6^	T/C	*FZD6*/intron 1–2

Chr—Chromosome; HC—Height at croup; WC—Width of chest; WB—Width of back; DAT—Depth of adipose tissue in lumbar region.

**Table 2 genes-14-01288-t002:** SNP-in-known-meat-production genes loci in Karachaevsky sheep.

Gene	SNP	Chr	Position	LW, *p*-Value	DME, *p*-Value
*MSTN*	oar3_OAR2_118149265	2	118,149,265	N/A	N/A
*MSTN*	oar3_OAR2_118150665	2	118,150,665	N/A	N/A
*MEF2B*	oar3_OAR5_3860373	5	3,860,373	0.459	0.305
*MEF2B*	oar3_OAR5_3867887	5	3,867,887	N/A	N/A
*FABP4*	oar3_OAR9_57537070	9	57,537,070	N/A	N/A
*GH*	oar3_OAR11_47529756	11	47,529,756	0.476	0.726
*GH*	oar3_OAR11_47545769	11	47,545,769	0.305	0.514
*REM1*	oar3_OAR13_60384593	13	60,384,593	0.398	0.642
*REM1*	oar3_OAR13_60385591	13	60,385,591	0.573	0.460
*MYOD1*	oar3_OAR15_3434222	15	3,434,222	0.741	0.758
*MYOD1*	oar3_OAR15_3441596	15	3,441,596	0.436	0.233
*FST*	OAR16_27849538.1	16	25,631,318	0.383	0.416
*FST*	oar3_OAR16_25632659	16	25,632,659	0.457	0.585
*FST*	oar3_OAR16_25632701	16	25,632,701	0.614	0.743
*FST*	oar3_OAR16_25633632	16	25,633,632	0.492	0.178
*FST*	oar3_OAR16_25638968	16	25,638,968	0.430	0.272
*DLK1*	oar3_OAR18_64313560	18	64,313,560	0.718	0.628
*DLK1*	oar3_OAR18_64314938	18	64,314,938	0.656	0.270
*DLK1*	oar3_OAR18_64341672	18	64,341,672	0.742	0.519

Chr—Chromosome, N/A—no SNPs detected, LW—Live weight, DME—Depth of muscle “eye”.

## Data Availability

Data are available from the corresponding author upon request.

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
