# Peer review of "Genome-Wide Search for Associations with Meat Production Parameters in Karachaevsky Sheep Breed Using the Illumina BeadChip 600 K"

_genes, 2023, doi:10.3390/genes14061288_

Round 1
Reviewer 1 Report
In the manuscript (MS), the authors describe SNP polymorphisms and candidate genes associated with values like height at croup, width of chest, width of back, and depth of adipose tissue in lumbar region in Karachaevsky sheep.
The authors also investigated the effect of genes described elsewhere whether they are associated with the measurements performed.
I suggest the MS revise several changes suggested below.
Suggestions, notices:
In the abstract, authors describe; ‘In known productivity genes (MSTN, MEF2B, FABP4, etc.) locuses were found a significant absence of influence on the dispersion of the Karachaevsky sheep phenotypes. This confirms their possible participation in the formation of the phenotypes of productivity parameters in animals…’.
I believe the word ‘their’ is for the presented candidate genes, not for ‘MSTN, MEF2B, FABP4, etc.’. If so, please reformat the sentence to make it clear.
On the other hand, I do not agree that the absence of ‘MSTN, MEF2B, FABP4, etc.’ among the candidate genes is a confirmation of what the authors describe here. The absence is just a fact here, as shown in Table 2., which is actually a very good idea to present in the MS.
Line 103: ‘…, and fed with a total mixed ration.’
It might be interesting to describe what the mixed rations composed of.
Line 117:
Instead of ‘Substitutions with a minor allele frequency (MAF) of less…’, use ‘Loci with a minor allele frequency (MAF) of less...’
Line 120: Please describe what is the ‘positive result’. Or omit that phrase. Or just describe, this amount of samples remained to be studied after quality control.
Line 124: In that section I miss, how the phenotypic values and the animals were organised into the analysis. Were the phenotypic values (PVs) treated as continuous variables? Or animals were organised into two groups having high and low values of PVs? Were PVs used as is, or were PVs recoded/standardised somehow?
Line 135:’Gene annotation…’
Gene annotation is a process during the deposit of a sequence or sequences to a database. As I see the authors used the browsers for retrieving information. Please rephrase that sentence.
Line 136-137: More readable in that form: ‘A significant difference was accepted if p-value was lower than 0.01.’
Line 142: ’…-log10(p) = 5’
In that equation p is equal to 0.00001, which is not the value described at Line 137. Please write something in the text to resolve/clarify that issue.
Figure 2.: Please define what the horizontal axis is.
Line 210-212: Dividing animals into two or more groups, based on complex genotype, based on thousands of genotype, is possible. For example, have the authors tried to use only MSTN, MEF2B, FABP4, GH, REM1, MYOD1, and FST polymorphism alone in a principal component analysis?
Please rephrase that ‘Due to the impossibility…’ sentence! If possible, the authors might present the result of the PCA mentioned above, especially when the animals are divided into two or more groups.
Line 351: In the review process I have not seen that supplement file. Please do not forget to include that. By the way, please specify what it contains.
Author Response
Dear Editor,
I accept all recommended changes from my respective reviewers. Thank them for their work, it will improve our manuscript for much better.
Regards,
Alexander Krivoruchko, corresponding author.
Reviewer 1.
In the manuscript (MS), the authors describe SNP polymorphisms and candidate genes associated with values like height at croup, width of chest, width of back, and depth of adipose tissue in lumbar region in Karachaevsky sheep.
The authors also investigated the effect of genes described elsewhere whether they are associated with the measurements performed.
I suggest the MS revise several changes suggested below.
Suggestions, notices:
In the abstract, authors describe; ‘In known productivity genes (MSTN, MEF2B, FABP4, etc.) locuses were found a significant absence of influence on the dispersion of the Karachaevsky sheep phenotypes. This confirms their possible participation in the formation of the phenotypes of productivity parameters in animals…’.
I believe the word ‘their’ is for the presented candidate genes, not for ‘MSTN, MEF2B, FABP4, etc.’. If so, please reformat the sentence to make it clear.
On the other hand, I do not agree that the absence of ‘MSTN, MEF2B, FABP4, etc.’ among the candidate genes is a confirmation of what the authors describe here. The absence is just a fact here, as shown in Table 2., which is actually a very good idea to present in the MS.
Rephrased: “Our study confirms the possible involvement of the identified candidate genes in the formation of the phenotypes…”
Line 103: ‘…, and fed with a total mixed ration.’
It might be interesting to describe what the mixed rations composed of.
Added: “From April to September, the sheep were grazing in the alpine meadows of the Caucasus. During the rest of the year, they were kept in the lowlands fed with hay and concentrates in a ratio of 10:1.”
Line 117:
Instead of ‘Substitutions with a minor allele frequency (MAF) of less…’, use ‘Loci with a minor allele frequency (MAF) of less...’
Changed to: “Loci with a minor allele frequency (MAF) of less”
Line 120: Please describe what is the ‘positive result’. Or omit that phrase. Or just describe, this amount of samples remained to be studied after quality control.
Rephrased: “From all 276 genotyped animals, 271 samples underwent genotyping quality control and was studied in the second stage.”
Line 124: In that section I miss, how the phenotypic values and the animals were organised into the analysis. Were the phenotypic values (PVs) treated as continuous variables? Or animals were organised into two groups having high and low values of PVs? Were PVs used as is, or were PVs recoded/standardised somehow?
When performing GWAS, we used quantitative parameters for each feature. Plink software allows you to perform associative analysis for both qualitative traits (then animals are divided into two groups) and quantitative traits (continuous variables). In this case, --qassoc is used instead of the --assoc operator.
Added: “The genome-wide association study was performed using the PLINK V.1.07 software [20] for quantitative traits.”
Line 135:’Gene annotation…’
Gene annotation is a process during the deposit of a sequence or sequences to a database. As I see the authors used the browsers for retrieving information. Please rephrase that sentence.
Changed to: “Description of genes…”
Line 136-137: More readable in that form: ‘A significant difference was accepted if p-value was lower than 0.01.’
Changed to: “A significant difference was accepted if p-value was lower than 0.01.” for individual SNP and lower 0.00001 for GWAS.”
Line 142: ’…-log10(p) = 5’
In that equation p is equal to 0.00001, which is not the value described at Line 137. Please write something in the text to resolve/clarify that issue.
Changed to: “…0.01 for individual SNP and lower 0.00001 for GWAS.”
Figure 2.: Please define what the horizontal axis is.
Added: “Clustering by individual genotypes located along the x-axis”
Line 210-212: Dividing animals into two or more groups, based on complex genotype, based on thousands of genotype, is possible. For example, have the authors tried to use only MSTN, MEF2B, FABP4, GH, REM1, MYOD1, and FST polymorphism alone in a principal component analysis?
Please rephrase that ‘Due to the impossibility…’ sentence! If possible, the authors might present the result of the PCA mentioned above, especially when the animals are divided into two or more groups.
We meant that the division of animals into two groups is impossible specifically in our samples set. This is indicated by clustering into several groups at once already at the first and second levels.
We did not perform PCA for polymorphisms in known productivity genes, since it was important for us to determine the possibility of individual influence on phenotype parameters in Karachaevsky sheep.
Rephrased: “If we cannot to divide animals into two groups based on a complex genotype, then assessment of its effect on the parameters of meat productivity of animals was not performed.”
Line 351: In the review process I have not seen that supplement file. Please do not forget to include that. By the way, please specify what it contains.
Changed: “Data are available from the corresponding author upon request”

Reviewer 2 Report
1. In the research background, it is necessary to introduce Karachaevsky sheep breed.
2. The genes on line 15, 16 and 189 should be italicized.
3. In line 131-132, candidate genes was performed in half of a centimorgans region, (250,000 bp upstream and downstream ), What is the basis for your choice? Or you can list the cited references.

Author Response
Dear Editor,
I accept all recommended changes from my respective reviewers. Thank them for their work, it will improve our manuscript for much better.
Regards,
Alexander Krivoruchko, corresponding author.
Reviewer 2.
This paper has great reference value for improving sheep meat production
- In the research background, it is necessary to introduce Karachaevsky sheep breed.
The description of the breed is given in the Introduction. Added: «The researchers noted the independent origin of the breed by improving local sheep and the absence of genetic links with modern sheep breeds.. … The breed is well adapted for breeding in year-round grazing conditions. … Today, the farms of Karachay-Cherkessia contain more than 250,000 heads of this breed. …. The meat has a high taste … The fat content of milk is on average 9.6%»
- The genes on line 15, 16 and 189 should be italicized.
Genes in these Lines was italized.
- In line 131-132, candidate genes was performed in half of a centimorgans region, (250,000 bp upstream and downstream ), What is the basis for your choice? Or you can list the cited references.
Added: “Within this interval, the frequency of crossingover is only 0.5%, and the SNP is highly likely to be inherited together with the adjacent gene.”
